# A Comprehensive Integrated Genetic Map of the Complete Karyotype of *Solea senegalensis* (Kaup 1858)

**DOI:** 10.3390/genes12010049

**Published:** 2020-12-31

**Authors:** Manuel A. Merlo, Silvia Portela-Bens, María E. Rodríguez, Aglaya García-Angulo, Ismael Cross, Alberto Arias-Pérez, Emilio García, Laureana Rebordinos

**Affiliations:** Genetics Area, Faculty of Marine and Environmental Sciences, INMAR, University of Cadiz, 11510 Cádiz, Spain; alejandro.merlo@uca.es (M.A.M.); silvia.portela@uca.es (S.P.-B.); mariaesther.rodriguez@uca.es (M.E.R.); aglaya.garcia@uca.es (A.G.-A.); ismael.cross@uca.es (I.C.); alberto.arias@uca.es (A.A.-P.); emiliomanuel.garcia@uca.es (E.G.)

**Keywords:** *Solea senegalensis*, cytogenetic map, BAC-FISH, physical map, comparative genomics

## Abstract

*Solea senegalensis* aquaculture production has experienced a great increase in the last decade and, consequently, the genome knowledge of the species is gaining attention. In this sense, obtaining a high-density genome mapping of the species could offer clues to the aquaculture improvement in those aspects not resolved so far. In the present article, a review and new processed data have allowed to obtain a high-density BAC-based cytogenetic map of *S. senegalensis* beside the analysis of the sequences of such BAC clones to achieve integrative data. A total of 93 BAC clones were used to localize the chromosome complement of the species and 588 genes were annotated, thus almost reaching the 2.5% of the *S. senegalensis* genome sequences. As a result, important data about its genome organization and evolution were obtained, such as the lesser gene density of the large metacentric pair compared with the other metacentric chromosomes, which supports the theory of a sex proto-chromosome pair. In addition, chromosomes with a high number of linked genes that are conserved, even in distant species, were detected. This kind of result widens the knowledge of this species’ chromosome dynamics and evolution.

## 1. Introduction

Since the 1970s and 1980s, aquaculture production has been progressively increasing until it became the fastest growing sector of the agrifood industry. This growth is the response to the demand for animal protein, functional foods and the need to alleviate the fishing grounds that are increasingly difficult to recover. As a final result, aquaculture production is now practically at the same level as fish production [1]. However, in order to achieve this challenge, it is not only important to increase the production of aquaculture species, but also to diversify cultivable species. In this sense, the Senegalese sole (*S. senegalensis*, Kaup 1858) has positioned itself as one of the priority species for the diversification of aquaculture in the southwestern region of Europe, due to its high market value and the qualities of its appreciated meat [2].

This has led several studies to deepen the knowledge of the biology of the Senegalese sole in order to translate it into an improvement for production. Thus, studies related to the life cycle, reproduction, physiology, pathology, genetics and genomics have been carried out [3,4,5,6,7]. However, the genomic data of this species are still scarce compared to other similar species and insufficient to be used in breeding programs.

Research in the field of genomics, in particular the study of genetic maps, is a very useful tool to provide a basis for the development of breeding programs [8,9,10]. Hence, genetic maps are essential tools, mainly in non-model species with limited genomic resources available, because they are useful for mapping quantitative trait loci in marker-assisted selection, positional cloning and genome assembly. Most genetic maps are based on microsatellites (SSRS), single nucleotide polymorphisms (SNPs), expressed sequence tags (EST) and bacterial artificial chromosomes (BACs). In aquaculture fish species, genetic maps have been constructed in a few species, starting with the fresh water ones, such as Atlantic salmon [11,12], Nile tilapia [13], rainbow trout [14], common carp [15,16] and catfish [17,18], and afterwards, marine fish like gilthead sea bream [10], Chinese tongue sole [19], turbot [20] and seabass [21].

Cytogenetic maps, in particular, consist of the physical location of regions of the DNA of the target species within the chromosomal complement of the species. This type of map has proven to be useful to complete the data of other types of maps, integrating both types of information. There are different works in which cytogenetic information has been used to integrate information from different maps, for example those carried out in *Salmo salar* [22], *Oncorhynchus mykiss* [23] and *Scophthalmus maximus* [20].

Mapping using BAC-FISH probes allows, in a straightforward and easy way, the final assembly and mapping of the genes in their chromosomes. BAC clones contain large inserts and they allow the localization of multiple probes for each chromosome, thereby raising the number of markers per chromosome karyotype and in each species. From these clones that are positioned in a library, the inserts can be sequenced with next generation sequencing techniques and, being used as probes, provide an efficient approach for anchoring linkage and genomic data in the physical chromosomes [24]. Integration of data at different scales, i.e., a genetic map with a physical map, might help gain new perspectives and allow a more comprehensive picture that is essential for comparative genome analysis [25].

For the sole species, linkage and physical maps have been described [24,26,27,28] and, currently, more than seven thousand scaffolds have been obtained [29]. During the last few years, great progress has been made in the genomics of Senegalese sole, with the transcriptome of the species being obtained in 2014 [30]. Regarding the cytogenetic map of this species, it started with the location of the multigenic families of 45S rDNA and 5S rDNA, in addition to the GATA repeats and telomeric sequences [31]. Later, it was found that the multigenic family of 5S rDNA was linked to different members of the multigenic families of snDNA [32]. Through the FISH-BAC technique, the acquisition of the cytogenetic map of the species has been deepened and integrated with physical maps, by using different genes of interest for the cultivation of the species. Similarly, it has been possible to characterize and locate the BAC clones with other types of sequences of interest, such as miRNAs or transposons [6,28,33,34]. As a result of these studies, evidence has accumulated that the pair of major metacentric chromosomes of *S. senegalensis* could be a pair of sex proto-chromosomes, which have followed a specific evolutionary pattern involving, among other events, Robertsonian fusion [33,35,36,37].

In view of this accumulation of data on the cytogenetic and physical map of the species, it is necessary to integrate them to deepen the structural genomics knowledge of Senegalese sole and to serve as a reference for further studies on the species, as well as in comparative genomics studies with other related species. Therefore, the objectives of this work is, firstly, to get a wide set of chromosomal markers in order to elaborate a high-density cytogenetic map, and secondly, to gain clues for deciphering the important key aspects to the sexual determining region, conserved gene regions as well as the localization and characterization of genes of interest for the culture improvement of the Senegalese sole. For these reasons, all cytogenetic and sequence data obtained in the species along with other unpublished data were reviewed and unified into a single high-density cytogenetic map and related to the sequence information. In this map, a high number of BAC clones were located (93), with 588 genes annotated, and it is estimated that it covers 2.43% of the genome of the species.

## 2. Materials and Methods

### 2.1. BAC Sequencing and Annotation

BAC clones were isolated from a *S. senegalensis* BAC library, composed of 29,184 positive clones, distributed in seventy-six 384-well plates. To isolate the BAC clones, including the genes of interest, specific primers and a 4D PCR method were used. Anonymous clones were randomly isolated from the genome. Specific primers were designed from Senegalese sole sequences available in the *S. senegalensis* database (SoleaDB, http://www.scbi.uma.es/soleadb) and using orthologous sequences from different fish species deposited in the ENSEMBL database (https://www.ensembl.org/index.html).

PCR was performed in a total volume of 25 µL containing 100 ng of the BAC-DNA template, 3 mM of MgCl_2_, 1 mM of each dNTP, 5 µmol of each primer and 1.5 U of NZYTaq™ DNA Polymerase (NZYTech, Lisbon, Portugal). The PCR conditions involved a first denaturing step at 95 °C for 2 min, followed by 30 cycles of 15 s at 95 °C, 15 s at 60 °C or 65 °C and 30 s at 72 °C.

After the BAC library screening, the positive BAC clones were isolated using the Qiagen Large-Construct Kit (Qiagen, Venlo, The Netherlands), which recovers large DNA quantities, following the manufacturer’s recommendations.

The BAC DNA was quantified by spectrophotometry using a Nanodrop^©^ (Thermo, Waltham, MA, USA). A yield of 5–12 µg was required for sequencing, which was done by Lifesequencing™ (Lifesequencing, Valencia, Spain) using 454 Technology (Roche, Basel, Switzerland) powered by the Genome Sequencing FLX System, or alternatively sequenced on the MiSeq Illumina sequencing platform (300 cycles of paired end reads, Lifesequencing, Valencia, Spain). The reads were assembled de novo using SPAdes software, version 3.11.1. (CAB SPbU, St Petersburg, Russia) The functional and structural annotation of the BAC sequences were analyzed in a semi-automatic process. Protein and EST from *S. senegalensis* and other fish species, such as *Danio rerio*, were compared. The homologous sequences obtained were used to get the best predictions. To localize the non-transcribed elements in the genome, RNA structure prediction tools were applied. Finally, all available information was used to create plausible models and, whenever possible, functional information was added. Using the genomic editor Apollo [38], Signal map software (Roche Applied Science, Basel, Switzerland) and Geneious Basic 5.6.5 (Biomatters Ltd., Auckland, New Zealand), the results were individually tested and adjusted in the final edition process. Alternatively, annotation was conducted as follows: Eukaryotic genes were predicted with Augustus v.3.3.3 guided by the homologous proteins of *D. rerio* and verified with the transcriptome of *S. senegalensis*. Then, the predicted ORFs (open reading frames) were functionally annotated with Blast2GO v1.1.135. With this procedure, the sequence genetic information (gene name, strand, position, etc.) was stored in gff format.

### 2.2. FISH Analysis

Chromosome preparations were made from *S. senegalensis* larvae aged 1–3 days. The specimens were pretreated with 0.02% colchicine for 3 h, subjected to hypotonic shock, and finally fixed in a freshly made Carnoy solution (absolute ethanol:acetic acid (3:1)). Larvae were homogenized in Carnoy and the preparations were dropped onto wet slides by splashing on a hot plate with damp paper.

Positive BAC clones were isolated using the Qiagen Plasmid Midi Kit (Qiagen, Venlo, The Netherlands), following the manufacturer’s recommendations. For double-FISH experiments, the BAC clones were labeled by nick translation using the DIG- or BIO-Nick translation Mix (Roche Molecular Biochemicals, Basel, Switzerland). For multiple-FISH experiments, the labeling was done with a first amplification by DOP-PCR followed by a conventional PCR for labeling, as described previously in Liehr [39]. Three different fluorochromes were used, i.e., Texas red (Thermo Fisher Scientific, Waltham, MA, USA), fluorescein-isothiocyanate (FITC) (Enzo, New York, NY, USA) and diethyl-aminocoumarin (DEAC) (Vysis, Shirley, NY, USA). Finally, the probes were precipitated using a standard protocol with sodium acetate and ethanol.

Chromosome preparations were pre-treated with pepsin solution at 37 °C and fixed with a paraformaldehyde solution. Finally, the preparations were dehydrated with an ethanol series of 70%, 90% and 100% and air-dried before hybridization. The processes of hybridization, post-hybridization and staining were carried out following the protocol described by García-Cegarra et al. [24] for double-FISH and by Portela-Bens et al. [35] for multiple-FISH. Double-FISH images were obtained using the fluorescence microscope Zeiss PALM MicroBeam equipped with an AxioCam MRm digital camera. Alternatively, multiple-FISH images were obtained with a digital CCD camera (Olympus DP70) coupled to a fluorescence microscope (Olympus BX51 and/or Zeiss Axioplan using software of MetaSystems, Altlussheim, Germany).

### 2.3. Synteny Analysis

Cross-species comparisons were carried out using CIRCOS software [40]. The comparison was performed from the ENSEMBL and NCBI database of three species, of which two are closely related, namely, tongue sole (*Cynoglossus semilaevis*) and turbot (*S. maximus*), and another one from a distant taxon, zebrafish (*D. rerio*).

## 3. Results

### 3.1. Chromosomal Map

Ninety-three BAC probes, distributed in 20 of the 21 chromosome pairs of the Senegalese sole (Figure 1 and Figure 2), were used. The chromosome where more BAC probes were detected was the largest metacentric chromosome (14 probes), followed by the metacentric 2 pair (9 probes); while, in the telomeric 14 pair, no homologous region was found to hybridize, and only one probe was found in the metacentric 3, submetacentric 5, telomeric 18 and telomeric 20 chromosomes. The relative position within each chromosome based on the FISH observations of these probes was 18 in the telomeric position, 17 in the subtelomeric position, 17 in the interstitial position, 23 in the subcentric position, 2 in the pericentromeric position and 1 in the centromeric position.

Fourteen BAC clones showed positive signals on more than one chromosome pair (Appendix A). For six of them (3F15, 19K18, 19L16, 25P16, 46P22 and 53I12), it was impossible to clearly differentiate the stronger signal corresponding to the homologous region of the BAC insert. Telomeric chromosome 14 did not display unique or main signals, but secondary ones; therefore, these are not the true homologous regions of the insert of those BAC clones. For other nine BAC probes (45M23, 46K16, 53I12, 54F6, 57G6, 59B23, 60P24, 71J13 and 73J17), the specific chromosome where they hybridized could not be determined; however, all of them hybridized to telomeric pairs (Figure 3). Two other BAC probes provided a scattered signal (Appendix A).

### 3.2. BAC Sequence and Annotation Data

After sequencing the 93 BAC clones used to hybridize, a total of 14.9 Mb of the genome was obtained, which represents about 2.43% coverage of the sole genome, assuming a total sole genome size of 612.3 Mb, as described by Manchado et al. [29].

A total of 588 genes were annotated (Appendix A), of which 522 could be positioned in their corresponding chromosomes by FISH (Table 1). The chromosome pair with the highest gene density was chromosome 3, followed by chromosome 2. It was also possible to observe that the average gene density of the metacentric chromosomes was considerably higher than that of the submetacentric (SMT), subtelocentric (STL) and telocentric (TL) chromosomes (Figure 4).

Some BAC clones overlapped, evidencing co-location (Appendix A). Thus, BAC clones 5K5, 10K23, 10L10 and 73B7 co-located, as well as BAC clones 11O20 and 20D18, 16E16 and 48K7, 45L11 and 4E10, 54F6 and 59B23, as well as 65I16 and 19K18.

### 3.3. Comparative Genomic Analysis

Comparative analysis by Circos diagrams evidenced more thick ribbons between *S. senegalensis* and *D. rerio* than between *S. senegalensis* and any of the other flatfish species (Figure 5). More chromosomal rearrangements were detected between *S. senegalensis* and *C. semilaevis* than between *S. senegalensis* and *S. maximus*.

The comparison between *S. senegalensis* and *D. rerio* also allowed us to identify large groups of linked and conserved genes between the species (wide ribbons). Such are the cases represented by chromosomes 1 (red ribbon), 4 (blue ribbon) and 16 (light pink ribbon) of *S. senegalensis*. The conserved linked genes of each of these chromosomes are represented in Appendix A in addition to the Gene Ontology (GO) links. The conserved group of the linked genes in chromosome 1 shared some common functions, such as tissue development (especially of muscle and osseous tissue), cellular transport, communication and response. Some genes on chromosome 4 shared functions related to protein organization, transport, modification and degradation. Meanwhile, many linked genes of chromosome 16 presented functions related to immune response. All genes of chromosome 9 of *S. senegalensis* were orthologous to those of chromosome 15 of *C. semilaevis* and chromosome 13 of *S. maximus*. Similarly, almost all genes of chromosome 11 of *S. senegalensis* were orthologous to those of chromosomes 7 and 20 of *C. semilaevis* and *S. maximus*, respectively. Moreover, the gene orthologues from the large metacentric chromosome 1 of *S. senegalensis* were distributed mainly among chromosomes 3, 18, 19, 20 and Z of *C. semilaevis*; and among chromosomes 4, 7, 9, 21 and 22 of *S. maximus*. With respect to the second metacentric chromosome, the orthologues were distributed among chromosomes 1, 2 and 9 of *C. semilaevis*; and among chromosomes 2, 4 and 8 of *S. maximus*.

## 4. Discussion

One of the aims of this study was to obtain chromosomal markers for all the chromosomes of the Senegalese sole. Through the hybridization of 93 BAC probes, it has been possible to obtain a wide cytogenetic map of Senegalese sole. Seventy-eight BAC probes were assigned to specific chromosomes. These data are similar to those obtained in other works where BAC probes were used to establish a cytogenetic map, as in the case of *Passiflora edulis*, where 36 BAC clones were used [41], or *Saccharum spontaneum*, with 114 BAC clones [42]. It was possible to obtain chromosomal markers for all the chromosomes of Senegalese sole. Chromosomal markers have been used for various applications; for example, the integration of cytogenetic and genomic data in different species of cichlids allowed the detection of large conserved synthetic regions [43], or the consolidation of the genetic, physical and cytogenetic maps, in the scallop species *Patinopecten yassinoensis* [44]. In the specific case of flatfish, it could be an additional tool to solve the controversial taxonomy of the group.

The inserts containing the BAC clones correspond to a single region of the sole’s genome, so it was surprising to see how in up to 12 BAC clones more than one pair of chromosomes with signs of hybridization were obtained, thus indicating a 15.38% chance of duplicated regions. In most cases, a pair of chromosomes with more intense signals, corresponding to the true locus of the insert, was identified. However, in five cases, a signal more intense than the others could not be distinguished (Figure 1—red lines). The fact that positive signals are observed in more than one chromosome pair could be the result of ancestral imprints left by the teleost-specific genome duplication event (TGD). It is expected that the more similarity the duplicated genes retain, the more similar in intensity the multiple hybridization signals of these BAC clones will be, thus making it difficult to discriminate the true homologous region. After a complete genome duplication event, the duplicated genes may suffer two destinies, either being lost or remaining in the genome. In the event that they remain in the genome, these new genes may either maintain the function of the ancestral gene, acquire a subfunction or acquire a new function [45]. It is believed that approximately 15% of the paralogs generated by TGD have been maintained in the teleost genome [46], a percentage very similar of that of the number of duplicated FISH signals obtained in this study and which could explain the adaptive plasticity of this group. Recently, it has been described that the fixation of extra copies of de novo DNA methyl transferases (*dnmt3*) genes may have contributed to the diversification and adaptive plasticity of the teleost [47]. Teleosts, including *S. senegalensis*, have a high number of functional *dnmt3* paralogs where up to 6 *dnmt3* paralogs with different tissue and developmental specific expression patterns have been described [48].

A total of 14.9 Mb were sequenced among all the BAC clones included in this work, a considerable amount considering that the inserts of the sole BAC library have an average size of 285 kB [35], although it only accounts for 2.43% of the total genome size of 612.3 Mb described by Manchado et al. [29]. Of the 588 annotated genes, some of them were related to sex determination or reproduction and were discussed in a previous work [35]. The largest metacentric pair has been proposed as a sex proto-chromosome pair, due to different lines of evidence, such as the accumulation of genes related to sex [35,37], existence of Robertsonian fusion [33,36], evidence of recurrent chromosomal inversions [36,37] and accumulation of repetitive sequences [28,36]. Recently, an analysis of the 24 genes related to the immune system in Senegalese sole showed some tendency to group in a few chromosomes, which could provide an evolutionary advantage [6].

A considerable difference has been observed in the mean gene density values between the MT- and SMT/STL/TL-type chromosomes. In humans, a strong correlation between high gene densities and effective bottleneck time has been described in autosomes, suggesting that autosomes with a higher gene density have undergone greater purification selection than those with a low density [49]. However, the largest metacentric chromosome presented a gene density (41.22 genes/Mb) of almost half the value of the other two metacentric chromosomes (80.69 and 85.64 genes/Mb, respectively). In plants, sex determination regions (SDR) are characterized by a reduction in gene density, a high concentration of repetitive sequences and an absence of recombination [50]. In the case of the Senegalese sole, the lower gene density found could be related to the presence of sex determination regions. For example, the BAC clone 48K7, located in a subcentromeric position of the largest metacentric chromosome (Appendix A), contains the genes *dmrt1*, *dmrt2* and *dmrt3*. These genes are genes frequently associated with sex determination [51,52]. Recently, Cross et al. [7] described an internal duplication of two exons of the *dmrt1* gene of *S. senegalensis*. Such duplication events have often been associated with neofunctionalization of the gene product, which, in this case, the new function could be involved with sex determination. However, the neofunctionalization is not the unique evolutionary fate of a duplicated gene. The BAC clone 48K7 is also located in a region with a high concentration of repetitive elements [37], a characteristic often observed in the evolution of the sex chromosomes [53]. One of the first steps in the evolution of the sex chromosomes is the suppression of recombination, evolutionarily selected to avoid the overcrossing of sex determination genes [54]. The accumulation of repetitive sequences in regions where possible sex determination genes exist and the decrease in gene density on the metacentric chromosome 1 of *S. senegalensis* could be associated with a decrease in recombination on this chromosome. All of these characteristics together with the accumulation of chromosomal reorganizations observed in that chromosome [36,37] reinforce the theory that the largest pair of metacentric chromosomes are proto-sex chromosomes.

The Circos analysis evidenced major rearrangements between the two most distant species, *S. senegalensis* and *D. rerio*, because it presented more thick ribbons than the comparison with any of the other Pleuronectiformes species. However, three large regions of conserved gene linkage have been observed and some of these genes shared common functions. It has been postulated that gene activity does not occur randomly across the genome and clusters of neighboring genes have been observed [55]. The reason for such a co-expression pattern is the co-regulation of genes with similar biological functions [56], the reduction of gene expression noise [57] or a tandem duplication, forming a paralogue copy with similar functions [58]. Many of the genes analyzed by comparative genome analysis share common functionalities, which could explain the linkage conservation across the different species treated in this work, independently of the particular gene organization [55]. In a previous study, conserved gene surroundings were detected in *S. senegalensis* by microsynteny analysis in the *nanos3* and *amh* genes, the latter with a gene neighborhood with shared functions in sexual maturation and cell cycling [35]. Recently, some genes related to metamorphosis and development were found to be closely linked in gene clusters [59].

## 5. Conclusions

The analysis and integration of the cytogenetic and physical maps of *S. senegalensis* have led to a broad knowledge about its genome organization and evolution. Large genome rearrangements, such as Robertsonian fusions and chromosomal inversions, were demonstrated to drive the evolution of the major metacentric chromosome. In addition, cytogenetic evidence (a low gene density, accumulation of repetitive sequences and the presence of *dmrt1*, *dmrt2* and *dmrt3* genes) reinforce the hypothesis that the metacentric chromosomes of pair 1 are proto-sex chromosomes. Comparative analysis with other fish model species has demonstrated the existence of conserved gene linkages, which allows a better understanding of the genome organization in fish. Moreover, cytogenetic mapping has indicated the existence of putative duplications that could be related to the TGD.

The advances in the cytogenetic and physical mapping of *S. senegalensis* also opens up future work lines, such as the full integration of its cytogenetic, genetic and physical maps, a large-scale study of its karyotype evolution and facilitating the assembly of its ongoing genome project. In addition, this kind of data could help in resolving the taxonomic controversies around the Pleuronectiformes order.

## Figures and Tables

**Figure 1 genes-12-00049-f001:**
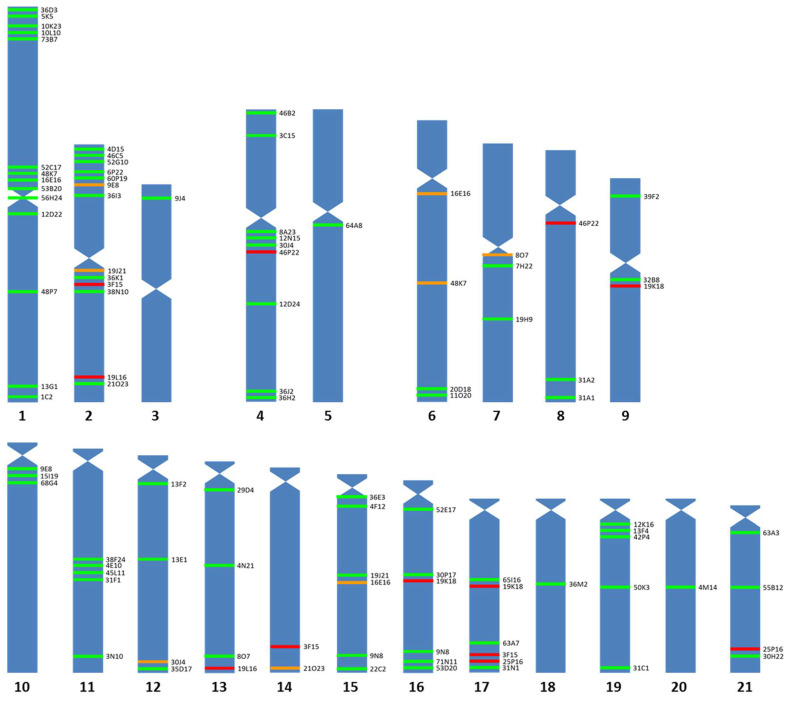
Idiogram of *Solea senegalensis* showing positive signals of 83 BAC probes. Seventy-eight of them were assigned to specific chromosomes (green lines) and five showed multiple signals that cannot be assigned to a specific chromosome (red lines). Orange lines represent the secondary signals of the specifically located BAC probes.

**Figure 2 genes-12-00049-f002:**
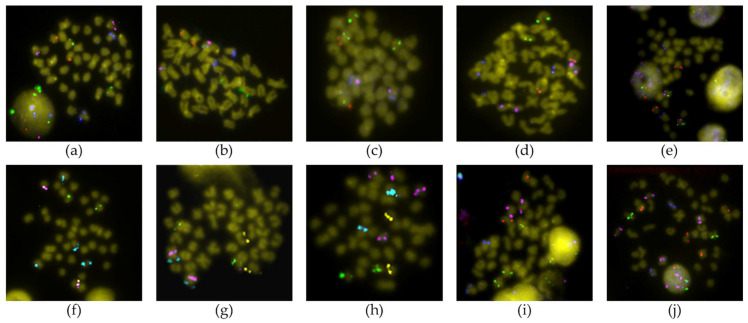
Representative BAC-FISH showing a marker in each chromosome of *Solea senegalensis*. (**a**) 4M14 (green)—15I19 (red)—22C2 (pink)—32B8 (blue); (**b**) 63A3 (green)—63A7 (red)—8O7 (pink)—9J4 (blue); (**c**) 19J21 (green)—6P22 (red)—7H22 (pink)—19H9 (blue); (**d**) 63A7 (green)—30H22 (red)—4N21 (pink)—13F4 (blue); (**e**) 4F12 (green)—36M2 (red)—36E3 (pink)—31A2 (blue); (**f**) 4N21 (green)—71N11 (yellow)—53D20 (pink)—30J4 (cyan); (**g**) 10L10 (green)—71N11 (yellow)—38N10 (pink)—52G10 (cyan); (**h**) 13E1 (green)—38F24 (yellow)—9E8 (pink)—4M14 (cyan); (**i**) 46P22 (green)—71N11 (red)—20D18 (pink)—19H9 (blue); (**j**) 29D4 (green)—21O23 (red)—30J4 (pink)—4F12 (blue).

**Figure 3 genes-12-00049-f003:**
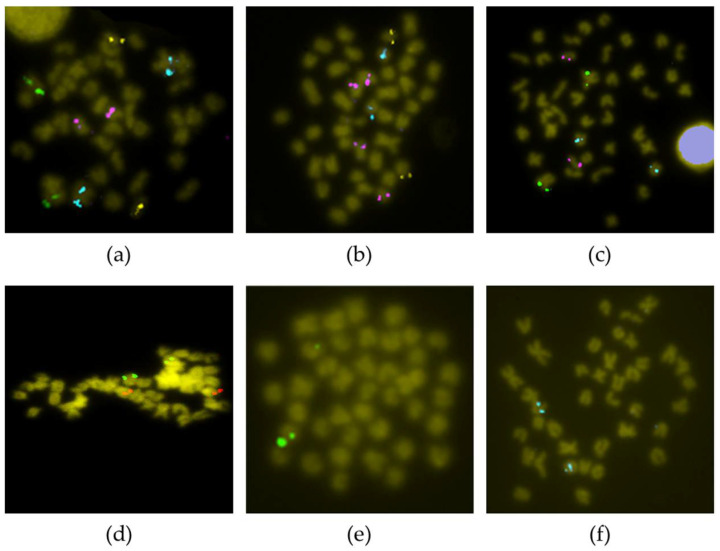
BAC-FISH with probes not assigned to a particular chromosome. (**a**) 54F6 (green)—45M23 (yellow)—53I12 (pink)—73B7 (cyan); (**b**) 73J17 (green)—46K16 (yellow)—53I12 (pink)—68G4 (cyan); (**c**) 73B7 (green)—57G6 (pink)—54F6 (cyan); (**d**) 71J13 (green)—53I12 (red); (**e**) 60P24 (green); (**f**) 59B23 (cyan).

**Figure 4 genes-12-00049-f004:**
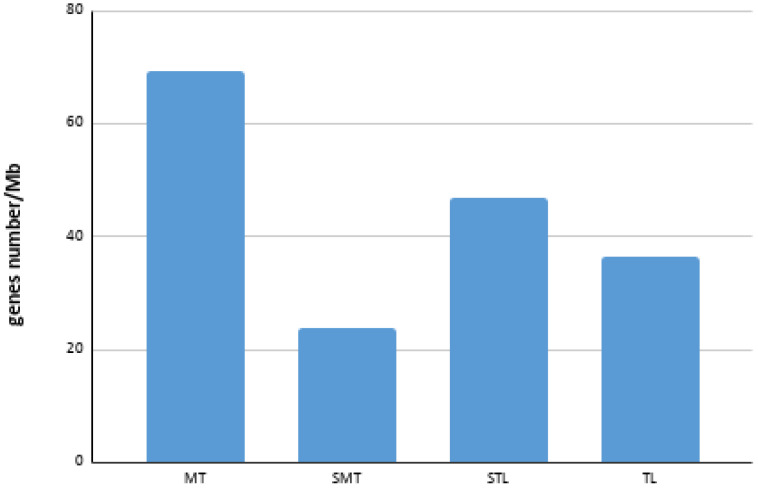
Bar chart of average gene density in each chromosome type. MT: metacentric; SMT: submetacentric; STL: subtelocentric; TL: telocentric.

**Figure 5 genes-12-00049-f005:**
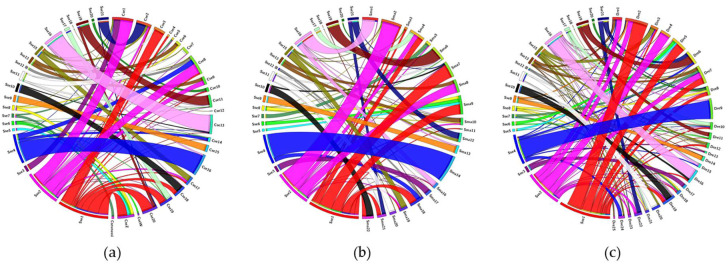
Circos diagrams between the BAC clones located on the chromosomes of *Solea senegalensis* and the orthologues on the chromosomes of (**a**) *Cynoglossus semilaevis*, (**b**) *Scophthalmus maximus* and (**c**) *Danio rerio*.

**Table 1 genes-12-00049-t001:** Gene density of each chromosome calculated from the number of genes annotated and the total length of all BAC contigs.

Chromosome	Number of BAC	Number of Genes ^a^	Size (bp) ^a^	Gene Density (Genes/Mb)
1	14	113	2,741,505	41.22
2	9	80	991,406	80.69
3	1	12	140,126	85.64
4	8	69	1,200,692	57.47
5	1	5	211,201	23.67
6	2	8	256,445	31.20
7	2	8	185,169	43.20
8	2	9	146,485	61.44
9	2	14	273,741	51.14
10	2	15	203,915	73.56
11	5	18	555,500	32.40
12	2	1	137,654	7.27
13	3	15	432,796	34.66
14				
15	3	21	431,021	48.72
16	5	40	583,480	58.56
17	3	11	448,777	24.51
18	1	1	81,656	12.25
19	5	25	758,436	32.96
20	1	5	283,462	17.64
21	3	24	413,134	58.09

^a^ The BAC clones 3F15, 9E8, 13E1, 19K18, 19L16, 22C2 and 25P16 were omitted, because they were sequenced together.

## Data Availability

The data presented in this study are available in supplementary material.

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
