# Peer review of "A Comprehensive Integrated Genetic Map of the Complete Karyotype of Solea senegalensis (Kaup 1858)"

_genes, 2020, doi:10.3390/genes12010049_

Round 1

Reviewer 1 Report

Overall Comments

This manuscript reports on a broad but diffuse BAC sequencing and chromosome mapping effort on Solea senegalensis. The random location of each clone ensures that the authors get coverage across all (this was a little unclear) of the species’ chromosomes allowing some investigation of genomic rearrangement. The BAC library approach does feel a little dated, that said it does allow identification of physical location of loci on the chromosomes which appears to be the focus of the work. The manuscript is generally reasonably written though there are enough places where the wording becomes clunky or difficult to follow that I would suggest outside editing.

My major critique of the manuscript is that it seems to lack focus until the conclusion. The abstract, introduction, and much of the discussion all describe a more general screen, and it tends to feel a little unclear what the authors are actually looking for. A major advantage to having information on physical locations of genes in a genome is being able to understand the incidence and possible consequences of genomic rearrangements. This is a point the authors seem to bring up towards the end of the manuscript, but it (or similar) could probably be baked in a little better.

Abstract

From line 18 starting at “Moreover” to the end of the abstract- It would probably help to have some description of the major finding of the work and what it means. It does feel like the abstract is missing its “discussion” of the study’s key results.

Introduction

Lines 63-70: I would argue that these two paragraphs distract from the introduction and might be better deleted. As it is the next paragraph provides a better written overview of existing genomic investigations for the species making these paragraphs redundant and a little distracting. The phrase “understanding genomes from different dimensions” sounds like something out of a sci-fi TV show and should probably be reworded if the authors choose to retain the paragraphs in.

Lines 83-89: I think this paragraph needs to be reworded to express what the authors were hoping to find using their sequenced BACs. This is the paragraph where the purpose of the study should be described and that is not coming across here.

Materials & Methods

Line 106 and 107: The authors should describe their approach to sequencing in the current manuscript rather than point to other work. Platform and sequencing depth are both reasonably important details.

Discussion

This is reasonably well written, but again my main critique is that it is unclear what the authors are looking for in their dataset at the beginning of the discussion. It seems to get more of a sense of direction at the halfway point where the authors move from describing findings to talking about chromosome organization.

One quibble at line 268: Neofunctionalization is just one of the possible evolutionary fates of a duplicated gene, and also not the most common. It’s probably worth adding that caveat.

Author Response

Reply to reviewer 1

Comments and Suggestions for Authors

Overall Comments

This manuscript reports on a broad but diffuse BAC sequencing and chromosome mapping effort on Solea senegalensis. The random location of each clone ensures that the authors get coverage across all (this was a little unclear) of the species’ chromosomes allowing some investigation of genomic rearrangement. The BAC library approach does feel a little dated, that said it does allow identification of physical location of loci on the chromosomes which appears to be the focus of the work. The manuscript is generally reasonably written though there are enough places where the wording becomes clunky or difficult to follow that I would suggest outside editing.

My major critique of the manuscript is that it seems to lack focus until the conclusion. The abstract, introduction, and much of the discussion all describe a more general screen, and it tends to feel a little unclear what the authors are actually looking for. A major advantage to having information on physical locations of genes in a genome is being able to understand the incidence and possible consequences of genomic rearrangements. This is a point the authors seem to bring up towards the end of the manuscript, but it (or similar) could probably be baked in a little better.

Abstract

From line 18 starting at “Moreover” to the end of the abstract- It would probably help to have some description of the major finding of the work and what it means. It does feel like the abstract is missing its “discussion” of the study’s key results.

The last part of the abstract has been re-written considering the reviewer recommendation. The abstract has been stated as follows: “Solea senegalensis aquaculture production has experienced a great increase in the last decade and, consequently, the genome knowledge of the species is gaining attention. In this sense, obtaining a high-density genome mapping of the species could offer clues to the aquaculture improvement in those aspects not resolved so far. In the present article, a review and new processed data have allowed to obtain a high-density BAC-based cytogenetic map of S. senegalensis beside the analysis of the sequences of such BAC clones to achieve integrative data. A total of 93 BAC clones were used to localize within the chromosome complement of the species and 588 genes were annotated, thus reaching almost the 2.5% of the S. senegalensis genome sequences. As a result, important data about the genome organization and evolution has been obtained, such as the lesser gene density of the large metacentric pair compared with the other metacentric chromosomes, which supports the theory of a sex proto-chromosome pair. In addition, it has been detected chromosomes with a high number of linked genes that are conserved even in distant species. This kind of results widens the knowledge of the chromosome dynamics and evolution.”

Introduction

Lines 63-70: I would argue that these two paragraphs distract from the introduction and might be better deleted. As it is the next paragraph provides a better written overview of existing genomic investigations for the species making these paragraphs redundant and a little distracting. The phrase “understanding genomes from different dimensions” sounds like something out of a sci-fi TV show and should probably be reworded if the authors choose to retain the paragraphs in.

The two paragraphs R1 refers to have been almost entirely deleted. The last sentence in the second paragraph has been moved tothe end of the previous paragraph (that of between the lines 56-62) and the word “dimensions” has been changed by “points of view”. The essential information of the first paragraph has been synthesized in a short sentence and included at the beginning of the next paragraph (that of between the lines 71-82). The numbers of the citations implied in such changes have been reordered in both, the text and the reference list.

Lines 83-89: I think this paragraph needs to be reworded to express what the authors were hoping to find using their sequenced BACs. This is the paragraph where the purpose of the study should be described and that is not coming across here.

The paragraph has been deeply re-written in order to clarify the aim of the study. The paragraph has been reworded as follows: “In view of this accumulation of data on the cytogenetic and physical map of the species, it is necessary to integrate them to deepen in the structural genomics knowledge of Senegalese sole and to serve as a reference for further studies on the species, as well as in comparative genomics studies with other related species. Therefore, the purposes of this work is, firstly, to get a wide set of chromosomal markers in order to elaborate a high-density cytogenetic map, and secondly, to gain clues for deciphering important key aspects as the sexual determining region, conserved gene regions, as well as the localization and characterization of genes of interest for the culture improvement of the Senegalese sole. For these reasons, all cytogenetic and sequence data obtained in the species along with other unpublished data have been reviewed and unified into a single high-density cytogenetic map and related to the sequence information. In this map, a high number of BAC clones were located (93), with 588 genes annotated and it is estimated that it covers the 2.43% of the genome of the species.”

Materials & Methods

Line 106 and 107: The authors should describe their approach to sequencing in the current manuscript rather than point to other work. Platform and sequencing depth are both reasonably important details.

Sequencing and annotation procedures have been detailed as follows: “The BAC DNA was quantified by spectrophotometry using a Nanodrop© (Thermo, USA). A yield of 5–12 µg was required for sequencing, which was done by Lifesequencing™ (Valencia, Spain) using 454 Technology (Roche, Switzerland) powered by the Genome Sequencing FLX System, or alternatively sequenced on the MiSeq Illumina sequencing platform (300 cycles of paired end reads). The reads were assembled de novo using SPAdes software, version 3.11.1. The functional and structural annotation of the BAC sequences was analyzed in a semi-automatic process. Protein and EST from S. senegalensis and other fish species, such as Danio rerio, were compared. The homologous sequences obtained were used to get the best predictions. To localize non-transcribed elements in the genome, RNA structure prediction tools were applied. Finally, all available information was used to create plausible models and, whenever possible, functional information was added. Using the genomic editor Apollo [39], Signal map software (Roche Applied Science, Switzerland), and Geneious Basic 5.6.5 (Biomatters Ltd.), the results were individually tested and adjusted in the final edition process. Alternatively, annotation was conducted as follows: Eukaryotic genes were predicted with Augustus v.3.3.3 guided by the homologous proteins of D. rerio and verified with the transcriptome of Solea senegalensis. Then, the predicted ORF (open reading frames) were functionally annotated with Blast2GO v1.1.135. With this procedure, sequence genetic information (gene name, strand, position...) was stored in gff format.”

Discussion

This is reasonably well written, but again my main critique is that it is unclear what the authors are looking for in their dataset at the beginning of the discussion. It seems to get more of a sense of direction at the halfway point where the authors move from describing findings to talking about chromosome organization.

Some comments have been added in the two first paragraphs of the discussion that clarify the sense of the discussion and align with the purposes already stated in the final paragraph of the introduction.

One quibble at line 268: Neofunctionalization is just one of the possible evolutionary fates of a duplicated gene, and also not the most common. It’s probably worth adding that caveat.

We have not been able to understand what R1 referred to with this remark since we have not mentioned neofunctionalization in line 268. We have supposed that R1 refers to the discussion between lines 274-277, and we have reworded them for clarity: “Recently, Cross et al. [7] described an internal duplication of two exons of the dmrt1 gene of S. senegalensis. Such duplication events have often been associated with neofunctionalization of the gene product, which, in this case, the new function could be involved with sex determination. However, the neofunctionalization is not the unique evolutionary fate of a duplicated gene.”

Reviewer 2 Report

Pleuronectiform species have been intensely studied in recent years as models to understand the molecular mechanisms underpinning the severe transformations occurring during metamorphosis. As a result genomic resources accumulated and have already been used to interpret various aspects of the flatfish biology and physiology. In general, fish genomics holds potential to support genetic improvement and breeding programmes in aquaculture.

Present paper is set in this frame, and addresses the structure of the genome of the Senegalese sole Solea senegalensis, an economically important flatfish species in aquaculture, especially in southwest Europe.

Aim of the work is to provide a review on the available cytogenetic data, to update this information with unpublished data, and to originate a single high-density cytogenetic map for the Senegalese sole. The FISH-BAC technique is applied resulting the physical map of ninety-three BAC probes on 20 of the 21 pairs of chromosomes of the target species. The results are discussed in the frame of the available genomic information for the species. Furthermore a comparative analysis with other flatfishes and with the zebrafish was conducted using CIRCOS software.

Overall the paper is clearly written and informative, it contributes a significant piece of information to the general topic stressing the relevance of applying a cytogenomic approach to the study of fish genomes.

In my opinion the paper deserves publication, I just have some minor comments.   

Minor comments:

Line 112 – “Análisis” please change to “Analysis”

Line 131 – Please add technical information about the microscope and image acquisition system.

Line 264 – “major” please change to “largest”

Lines 296 – 297 – I suggest to modify the sentence “there is evidence that this chromosome should be considered as a sexual proto-chromosome pair” to “cytogenetic evidences (low gene density, accumulation of repetitive sequences, presence of dmrt1, dmrt2 and dmrt3 genes) reinforce the hypothesis that the metacentric chromosomes of pair 1 are proto-sex chromosomes"

Figures

Figure 1 - the figure is too small making it difficult to read the names of BAC probes. Would it be possible to split the ideogram into multiple rows, and enlarge the size of chromosomes? I think that clarity would benefit of such a change.

References

General comments:

  • Please italicize the volume number
  • digital object identifier (DOI) should be added where available
  • cited journals should be abbreviated according to ISO 4 rules

Specific comments:

Line 327 – delete “(2010)”

Line 345 – italicize “ Cyprinus carpio”

Line 354 – italicize “Salmo salar”

Line 358 – italicize “Oncorhynchus mykiss”

Line 366 – italicize “Ictalurus punctatus”

Line 370 - Reference n. 19 is the same as ref. n. 10, please delete

Line 403 – “2019” in bold

Line 407 – “2016” in bold

Author Response

Reply to reviewer 2

Comments and Suggestions for Authors

Pleuronectiform species have been intensely studied in recent years as models to understand the molecular mechanisms underpinning the severe transformations occurring during metamorphosis. As a result genomic resources accumulated and have already been used to interpret various aspects of the flatfish biology and physiology. In general, fish genomics holds potential to support genetic improvement and breeding programmes in aquaculture.

Present paper is set in this frame, and addresses the structure of the genome of the Senegalese sole Solea senegalensis, an economically important flatfish species in aquaculture, especially in southwest Europe.

Aim of the work is to provide a review on the available cytogenetic data, to update this information with unpublished data, and to originate a single high-density cytogenetic map for the Senegalese sole. The FISH-BAC technique is applied resulting the physical map of ninety-three BAC probes on 20 of the 21 pairs of chromosomes of the target species. The results are discussed in the frame of the available genomic information for the species. Furthermore a comparative analysis with other flatfishes and with the zebrafish was conducted using CIRCOS software.

Overall the paper is clearly written and informative, it contributes a significant piece of information to the general topic stressing the relevance of applying a cytogenomic approach to the study of fish genomes.

In my opinion the paper deserves publication, I just have some minor comments.   

Minor comments:

Line 112 – “Análisis” please change to “Analysis”

The correction has been made.

Line 131 – Please add technical information about the microscope and image acquisition system.

The following information has been added to the paragraph: “Double-FISH images were obtained using the fluorescence microscope Zeiss PALM MicroBeam equipped with an AxioCam MRm digital camera. Alternatively, multiple FISH images were obtained with a digital CCD camera (Olympus DP70) coupled to a fluorescence microscope (Olympus BX51 and/or Zeiss Axioplan using software of MetaSystems, Altlussheim, Germany).

Line 264 – “major” please change to “largest”

We did not find the word “major” at line 264. We have assumed that the author refers to line 272, so the correction has been made.

Lines 296 – 297 – I suggest to modify the sentence “there is evidence that this chromosome should be considered as a sexual proto-chromosome pair” to “cytogenetic evidences (low gene density, accumulation of repetitive sequences, presence of dmrt1, dmrt2 and dmrt3 genes) reinforce the hypothesis that the metacentric chromosomes of pair 1 are proto-sex chromosomes"

The suggestion has been taken into account.

Figures

Figure 1 - the figure is too small making it difficult to read the names of BAC probes. Would it be possible to split the ideogram into multiple rows, and enlarge the size of chromosomes? I think that clarity would benefit of such a change.

Figure 1 has been re-edited and enlarged.

References

General comments:

    Please italicize the volume number

    digital object identifier (DOI) should be added where available

    cited journals should be abbreviated according to ISO 4 rules

Corrections have been made.

Specific comments:

Line 327 – delete “(2010)”

The correction has been made.

Line 345 – italicize “ Cyprinus carpio”

The correction has been made.

Line 354 – italicize “Salmo salar”

The correction has been made.

Line 358 – italicize “Oncorhynchus mykiss”

The correction has been made.

Line 366 – italicize “Ictalurus punctatus”

The correction has been made.

Line 370 - Reference n. 19 is the same as ref. n. 10, please delete

The correction has been made.

Line 403 – “2019” in bold

The correction has been made.

Line 407 – “2016” in bold

The correction has been made.
